



# Solver for Hydrologic Unstructured Domain (SHUD): Numerical modeling of watershed hydrology with the finite volume method

Lele Shu[1], Paul Ullrich[1], and Christopher Duffy[2]

[1]Department of Land, Air, and Water Resources, University of California, Davis, Davis, California 95616, USA
[2]Department of Civil and Environmental Engineering, Pennsylvania State University, University Park, Pennsylvania 16802, USA

**Correspondence:** Lele Shu (lele.shu@gmail.com)

**Abstract.** Hydrological modeling is an essential strategy for understanding natural flows, particularly where observations are lacking in either space or time, or where topographic roughness leads to a disconnect in the characteristic timescales of overland and groundwater flow. Consequently, significant opportunities remain for the development of extensible modeling systems that operate robustly across regions. Towards the development of such a robust hydrological modeling system, this
paper introduces the Solver for Hydrological Unstructured Domain (SHUD), an integrated multi-process, multi-scale, multi-timestep hydrological model, in which hydrological processes are fully coupled using the Finite Volume Method. The SHUD integrates overland flow, snow accumulation/melting, evapotranspiration, subsurface and groundwater flow, and river routing, while realistically capturing the physical processes in a watershed. The SHUD incorporates one-dimension unsaturated flow, two-dimension groundwater flow, and river channels connected with hillslopes via overland flow and baseflow.

This paper introduces the design of SHUD, from the conceptual and mathematical description of hydrological processes in a watershed to computational structures. To demonstrate and validate the model performance, we employ three hydrological experiments: the V-Catchment experiment, Vauclin's experiment, and a study of the Cache Creek Watershed in northern California, USA.

    Possible applications of then SHUD model include hydrological studies from the hillslope scale to regional scale, water
resource and stormwater management, and coupling research with related fields such as limnology, agriculture, geochemistry, geomorphology, water quality, and ecology, climatic and landuse change. In general, SHUD is a valuable scientific tool for any modeling task involving simulating and understanding the hydrological response.

**Nomenclature**

**Evapotranspiration Calculation**





| | | |
|---|---|---|
| $\Delta$ | Slope vapour pressure curve $[kPaC^{-1}]$ | |
| $\gamma$ | Psychrometric constant $[kPaC^{-1}]$ | |
| $\lambda$ | Latent heat of vaporization $[MJkg^{-1}]$ | |
| 25 | $\rho_a$ | Density of Air $[kgm^{-3}]$ |
| | $c_p$ | Specific heat at constant pressure $[MJkg^{-1}C^{-1}]$ |
| | $e_a$ | Actual vapour pressure $[kPa]$ |
| | $e_s$ | Saturation vapour pressure $[kPa]$ |
| | $G$ | Soil heat flux density $[MJm^{-2}s^{-1}]$ |
| 30 | $r_a$ | Aerodynamic resistance $[sm^{-1}]$ |
| | $r_s$ | Surface resistance of vegetation $[sm^{-1}]$ |
| | $R_n$ | Net radiation at the crop surface $[MJm^{-2}s^{-1}]$ |

**Hydrological metrics**

| | | |
|---|---|---|
| | $\alpha$ | van Genutchten soil parameter $[m^{-1}]$ |
| 35 | $\alpha_h$ | Horizontal macropore areal fraction $[m^2m^{-2}]$ |
| | $\alpha_{imp}$ | Impervious area fraction $[m^2m^{-2}]$ |
| | $\alpha_{veg}$ | Vegetation fraction $[m^2m^{-2}]$ |
| | $\alpha_v$ | Vertical macropore areal fraction $[m^2m^{-2}]$ |
| | $\bar{K}$ | Average conductivity $[ms^{-1}]$ |
| 40 | $\bar{y}$ | Effective height of overland flow between two adjacent cells $[m]$ |
| | $\beta$ | van Genutchten soil parameter $[-]$ |
| | $\beta_s$ | Soil moisture stress to evapotranspiration $[-]$ |
| | $\Delta t$ | Time interval between consequential time steps $[m]$ |
| | $\overline{y_{gw}}$ | Effective water height for groundwater flow calculation $[m]$ |
| 45 | $\overline{y_{sf}}$ | Effective water height for overland flow calculation $[m]$ |



$\psi$      Soil matrix potential head $[m]$

$\Theta$      Relative saturation ratio $[-]$

$\theta$      Soil moisture content $[m^3 m^{-3}]$

$\theta_r$      Residual soil moisture content $[m^3 m^{-3}]$

$\theta_s$      Porosity of soil $[m^3 m^{-3}]$

$\theta_{fc}$      The soil moisture content of field capacity $[m^3 m^{-3}]$

$A_c$      Area of a cell $[m^2]$

$A_r$      Area of river open water $[m^2]$

$b_g$      Effective height of groundwater flow between the river segment and hillslope cell $[m]$

$b_s$      Effective height of overland flow between the river segment and hillslope cell $[m]$

$C_{ic}$      Coefficient of interception $[m]$

$C_w$      Coefficient of discharge $[m]$

$d_j$      Distance between centroids of the current cell and neighbor $j$ $[m]$

$d_{rb}$      Thickness of river bed; for calculation of baseflow to rivers $[m]$

$D_{us}$      The deficit of soil column; thickness of vadose layer $[m]$

$E_0$      Potential evapotranspiration $[m s^{-1}]$

$E_c$      Evaporation from interception $[m s^{-1}]$

$E_{sm}$      Evaporation from the soil matrix $[m s^{-1}]$

$E_{sp}$      Evaporation from ponding water on land surface $[m s^{-1}]$

$E_s$      Evaporation from soil $[m s^{-1}]$

$E_{tg}$      Transpiration from saturated layer $[m s^{-1}]$

$E_t$      Transpiration $[m s^{-1}]$

$H_{cgw}$    Hydraulic head of water in cell groundwater $[m]$

$H_{csf}$    Hydraulic head of water on land surface $[m]$





$H_{riv}$ Hydraulic head in a river channel $[m]$

$k_m$ Saturated conductivity of soil macropore $[ms^{-1}]$

$K_r(\Theta)$ Relative conductivity, which is a function of saturation ratio $[-]$

$k_x$ Saturated conductivity of the top soil $[ms^{-1}]$

$K_{eg}$ Effective horizontal conductivity $[ms^{-1}]$

$K_{ei}(\Theta)$ Effective infiltration conductivity $[ms^{-1}]$

$K_{er}$ Effective recharge conductivity $[ms^{-1}]$

$k_g$ Saturated horizontal conductivity $[ms^{-1}]$

$k_{rb}$ Saturated conductivity of the river bed $[-]$

$k_v$ Saturated vertical conductivity of saturated layer $[ms^{-1}]$

$L_j$ Length of edge $j$ of a cell $[m]$

$L_s$ Length of river segment that overlay with a cell $[m]$

$LAI$ Leaf Area Index $[m^2m^{-2}]$

$m_f$ Snow melting factor $[ms^{-1}C^{-1}]$

$n$ Manning's roughness $[sm^{-1/3}]$

$N_c$ Number of cells overlaying a river reach $[-]$

$N_u$ Number of upstream reaches flowing to a river reach $[-]$

$P$ Atmospheric precipitation or irrigation $[ms^{-1}]$

$P_n$ Net precipitation $[ms^{-1}]$

$P_{sn}$ Snowfall $[ms^{-1}]$

$Q_{dn}$ Volume flux to the downstream river channel $[m^3s^{-1}]$

$Q_{gr}$ Volume flux between river and cells via groundwater flow $[m^3s^{-1}]$

$Q_g$ Groundwater flow between two cells $[m^3s^{-1}]$

$q_i$ Infiltration rate, positive is downward $[ms^{-1}]$



$q_r$    Recharge rate, positive is downward $[ms^{-1}]$

$q_{sn}$    Snow melting rate $[m^3s^{-1}]$

$Q_{sr}$    Volume flux between river and hillslope cells via overland flow $[m^3s^{-1}]$

$Q_s$    The overland flow between two cells $[m^3s^{-1}]$

$Q_{up}$    Volume flux from upstream river reaches $[m^3s^{-1}]$

$s_y$    Specific yield $[mm^{-1}]$

$s_0$    The slope of land surface $[mm^{-1}]$

$S_{ic}$    The interception on the canopy $[ms^{-1}]$

$S_{ic}$    Water storage of interception layer (canopy) $[m]$

$S_{ic}^*$    Maximum interception capacity $[m]$

$S_{sn}$    Snow storage $[m]$

$T$    Air temperature $[C]$

$T_0$    Temperature threshold for snowmelt to occur $[C]$

$y_{gw}$    Groundwater head (above impervious bedrock) of a cell $[m]$

$y_{riv}$    River stage in a river channel $[m]$

$y_{sf}$    Surface water storage in a cell $[m]$

$y_{us}$    Unsaturated storage equivalence of a cell $[m]$

$z_{bank}$    Elevation of the riverbank from the datum $[m]$

$z_b$    Elevation of impervious bedrock from the datum $[m]$

$z_{gw}$    Elevation of groundwater table from the datum $[m]$

$z_m$    Elevation of macropore from the datum $[m]$

$z_{sf}$    Elevation of land surface from the datum $[m]$

**Variables used in CVODE**

$\mathbf{Y_0}$    The initial conditions to start the simulation. $[m]$





$Y_{gw}$     Vector of cell groundwater head (above impervious bedrock) $[m]$

$Y_{riv}$     Vector of river stage in all river channels $[m]$

$Y_{sf}$     Vector of surface water storage of all cells $[m]$

$Y_{us}$     Vector of unsaturated storage equivalence of all cells $[m]$

$Y$     Vector of conserved state variables in CVODE $[m]$

$t$     Time $[s]$

$t_n$     Current time $[s]$

$t_{n-1}$     Previous time $[s]$

## 1 Introduction

Certain scientific and applied questions are difficult to address with available observational data, and hence extrapolation of these limited datasets is often needed. Modeling is one of the cheapest and physically-consistent methods to perform quantitative extrapolation to events or systems where we may only have proxy measurements. Models inevitably help us better

understand the history of a given system or make decisions regarding the future, whether those systems be socioeconomic, hydrological, or climatological. The datasets produced through modeling can assist with decisions on infrastructural planning, water resource management, flood protection, contamination mitigation, and other relevant concerns.

A common statistical aphorism states, "*all models are wrong, but some are useful*". Due to trade-offs that occur in light of model complexity, computational resources, time-performance, available observations, and the "selective wrongs" of the

perceptual-conceptual-mathematical model design, models inevitably cannot tell the "whole truth" of an entire system, everywhere and at any time. Consequently, ongoing efforts by scientists and model developers have led to better models that are converging towards the "truth" and can provide more details of the nature of the truth. Nonetheless, these designs often focused on a particular objective – e.g., models are generally suitable or limited to particular research areas, purposes, or data availability.

In hydrology, lumped models (Hawkins et al., 1985; Fleming, 2010; Bergström, 1992) are fast and stable tools for estimating the discharge in river gages, assuming reliable meteorological data and observed discharge available. Lumped models disregard the spatial heterogeneity of terrestrial characteristics, instead of regarding the watershed as one unit based on statistical methods. Consequently, they are highly dependent on data availability and fidelity (Moradkhani and Sorooshian, 2008). Further, they rarely provide essential spatial metrics (e.g., soil moisture, groundwater, and evapotranspiration), and their parameters

lack definite physical meaning, which makes it challenging to interpret watershed characteristics or transfer parameters to other regions. On the other hand, distributed models (Beven, 2012; Lin et al., 2018; Gochis et al., 2015; Santhi et al., 2006;





Liang et al., 1996; Vivoni et al., 2011; Refsgaard et al., 1998; Shen and Phanikumar, 2010) are not perfect for all purposes either. The first challenge for distributed models is addressing complications and poor performance for large and high-resolution study regions. Although the model parameters, input, and output variables are spatially distributed, the conceptual descriptions

of the basic unit, such as Hydrological Representative Unit (HRU) in SWAT model, are of the lumped ideal. Further, models still use lumped calibration mode — that is, the "nudging" used in watershed calibration does not vary spatially as with the model configuration. Last but not least, the uncertainty from distributed forcing data and parameters is a big challenge within any distributed hydrological model (Beven, 2012; Blöschl et al., 2019) and others. Still, the development of new hydrological models has merit to leverage advances in mathematical and computing strategies, incorporate a fresh understanding of natural

processes, fix issues related to approximations or gaps in our understanding, and detect new outstanding issues.

Many successful hydrological models have been developed and are now available, providing significant and varied insight into water cycles from multiple perspectives (Beven, 2012). From the simplest lumped models (HEC-HMS (Fleming, 2010), HBV (Bergström, 1992)), to semi-distributed models (Beven, 1989; Beven and Germann, 1982; Beven and Kirkby, 1979), to complex distributed hydrological models (WRF-Hydro (Lin et al., 2018; Gochis et al., 2015), inHM (VanderKwaak, 1999),

PRMS (Leavesley et al., 1983), SWAT (Santhi et al., 2006), VIC (Liang et al., 1996), MIKE-SHE(Abbott and Refsgaard, 1996; Refsgaard et al., 1998), tRIBS (Vivoni et al., 2011, 2004, 2005) and PAWS (Shen and Phanikumar, 2010)), and even cutting-edge hydrological models based on machine-learning methods (Rasouli et al., 2012; Petty and Dhingra, 2018; Shen et al., 2018), all models have some distinctions and shortfalls related to performance, flexibility, and applicability.

Modelers, policymakers, and stakeholders have an ongoing and growing need for high-resolution and detailed information

about hydrological flows and the temporal-spatial distribution of water in a watershed. This need reflects the growing importance in coupling research with detailed long-term predictions and projections for ecological systems and the environment, agricultural development, and food security under future climate change. Global climate modeling, typically performed with a general circulation model, also requires information on soil moisture and groundwater fluctuations, which are related to streamflow and reservoir management (Hrachowitz and Clark, 2017; Blöschl et al., 2019).

The Solver for Hydrologic Unstructured Domain (SHUD) is a multi-process, multi-scale hydrological model where major hydrological processes are fully coupled using the Finite Volume Method (FVM). SHUD encapsulates the strategy for the synthesis of multi-state distributed hydrological models using the integral representation of the underlying physical process equations and state variables. As a heritage of Penn State Integrated Hydrologic Model (PIHM), the SHUD model is a continuation of 16 years of PIHM modeling in hydrology and related fields since the release of its first PIHM version (Qu,

175  2004).

The SHUD's design is based on a concise representation of a watershed and river basin's hydrodynamics, which allows for interactions among major physical processes operating simultaneously, but with the flexibility to add or drop states-processes-constitutive relations depending on the objectives of the numerical experiment for research purpose.

The SHUD is a distributed hydrological model in which the domain is discretized using an unstructured triangular irregular

network (e.g., Delaunay triangles) generated with constraints (geometric and parametric). A local prismatic control volume is formed by the vertical projection of the Delaunay triangles forming each layer of the model. Given a set of constraints (river





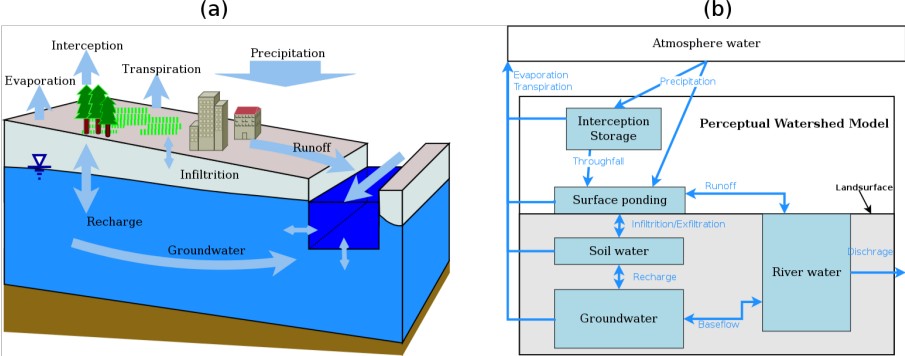

**Figure 1.** The conceptual schematic of hydrological processes in the SHUD model.

network, watershed boundary, elevation, and hydraulic properties), an "optimized mesh" is generated. The "optimized mesh" indicates the hydrological processes with the unstructured mesh can be calculated efficiently, stably and rationally (Farthing and Ogden, 2017; Vanderstraeten and Keunings, 1995; Kumar et al., 2009). River volume cells are also prismatic, with trapezoidal

or rectangular cross-section, and maintain the topological relation with the Delaunay triangles. The local control volumes encapsulate all equations to be solved and are herein referred to as the model kernel.

The objective of this paper is to introduce the design of SHUD, from the fundamental conceptual model of hydrology to governing hydrological equations in a watershed to computational structures describing hydrological processes. Section 2 describes the conceptual design and equations used in the model. In section 3, we employ three hydrological experiments to

demonstrate the simulation and capacity of the model. The three applications presented here are (1) the V-Catchment experiment, (2) the Vauclin experiment (Vauclin et al., 1979), and (3) the Cache Creek Watershed (CCW), a headwater catchment in Northern California. Section 4 summarizes the differences between SHUD and PIHM, then proposes possible applications of the SHUD model.

## 2   Model design

### 2.1   Conceptual description of hydrological system

We begin our introduction to the SHUD model with a conceptual description of water movement in a watershed (Fig. 1).

Surface and subsurface hydrology inevitably begins from atmospheric precipitation and other water inputs, including rainfall, snowfall and irrigation. Before precipitation reaches the land surface, it may first make contact with vegetation (e.g., leaves and branches). The water collected on vegetation above the land surface is referred to as canopy interception. When snow

is present, the snowfall accumulates on both land surface and within the canopy. All water (liquid and solid) staying on the canopy or the land surface is the total interception. When precipitation exceeds the interception capacity – the maximum water





that can stay on the canopy – the excessive precipitation falls to the land surface. The total water reaching the land surface is then called the net precipitation.

When precipitation reaches the land surface, water may flow horizontally and/or vertically. The horizontal flow is the surface runoff or overland flow along the terrestrial gradient that is relatively fast flow converging into steams. The vertical fluxes through the land surface include infiltration (for flow into the soil) and exfiltration (for flow out of the soil). Water that flows into the soil will then percolate to and raise the groundwater table. When the groundwater table reaches or exceeds the land surface, infiltration decreases to zero, and exfiltration may occur.

The water movement within the soil layer is usually slower than on the land surface. Vertically, the aquifer is divided into two layers based on its saturation status: the top unsaturated layer (or vadose layer) and saturated bottom layer (groundwater layer). These layers sit atop an assumed impermeable bedrock layer, an ordinary and reasonable approximation in hydrology that arises because of the relatively slow water exchange between shallow and deep confined aquifers, compared with fluxes between the land surface, river channel, and shallow groundwater. As horizontal unsaturated flow is relatively slow in the vadose layer compared with the vertical flow, it is a reasonable approximation to ignore the horizontal flow in the vadose layer when simulating at watershed scales. The downward vertical flow from the unsaturated layer to the saturated layer – groundwater recharge – is controlled by the soil moisture, soil characteristics and groundwater table. Positive recharge of groundwater increases the level of the groundwater table and reduces the thickness of the unsaturated layer. Within the saturated layer, the underground hydraulic gradient drives horizontal groundwater flow.

Runoff from the hillslope converges into river channels via surface runoff (runoff that travels overland to the stream channel) and baseflow (groundwater flow to a surface water body). However, water in the river channels could flow back to the hillslope when the river rises above its banks during flooding. The exchange of water between the river channels and groundwater is determined by the hydraulic gradient between the river stage and groundwater. Water in rivers flows downward until it exits the watershed.

Evaporation produces water loss from the canopy, land surface and soil, and consists of four components: evapotranspiration (ET) from interception storage, ponding water, soil moisture and groundwater. Transpiration occurs only when vegetation is present and could draw from the saturated groundwater when the groundwater level is high enough. Direct evaporation draws from interception, ponding water, and soil moisture.

Following the above description, several assumptions and simplifications are made in the SHUD model:

- The watershed is a closed domain, in which precipitation and discharge is the major flux into and out of the domain. This assumption is generally reasonable for most hydrological studies because both the lateral water flow from outside of the domain and water flux between the shallow groundwater and deep groundwater is minimal and insignificant to the water fluxes and mass balance. In a watershed where these fluxes are necessary, a modeler can modify the configuration of lateral boundary conditions to realize complex lateral fluxes.

- The horizontal flow within the unsaturated layer is zero. The hydraulic gradient in the vertical direction in the soil column is controlled by gravity leading to large vertical gradients of soil moisture content, whereas the horizontal gradient is





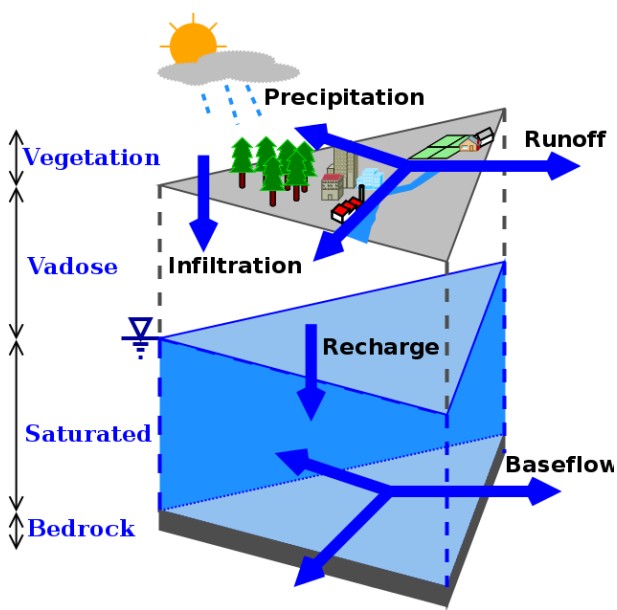

**Figure 2.** The three layers of the SHUD model and fluxes between layers.

smaller than the vertical one in magnitude. This assumption is invalid for microscale soil water movement but useful when the model grid spacing is from meters to kilometers (Beven, 2012).

– The evaporative fluxes that occur due to ET from rivers are ignored. Because the area of rivers exposed to the atmosphere is relatively small within a watershed, it is a reasonable approximation to lump the contribution of ET from the open water into the ET of the hillslope.


– The hydrological characteristics, including all physical parameters in soil, landuse, and terrain, are homogeneous within each cell. This is a common assumption in any distributed models, as the various models still need discretized domains instead of a continuous space. The next subsection elaborates on the parameters in each category.

– All geographic and hydraulic parameters do not change in time.


– Finally, SHUD uses a simplified representation of the geometry of the river networks due to the limitation of such data. This assumption is made because of the inherent challenges in measuring the geometry of the river cross-section everywhere along with the stream network.

## 2.2 Mathematical structure

The notation used in this section is summarized in list of symbols.


Figure 2 depicts the geometric structure of the discrete cells in SHUD. The watershed domain is discretized using an irregular unstructured triangular network (Delaunay triangles) generated with imposed spatial constraints. A prismatic control





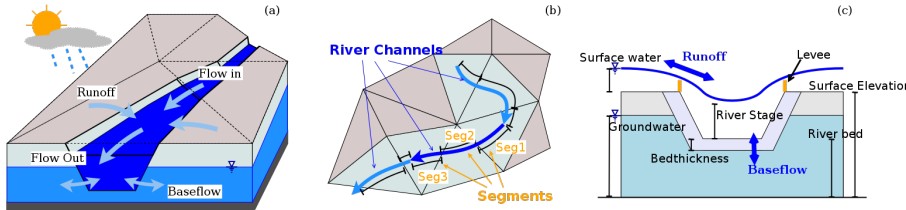

**Figure 3.** A depiction of the interaction between cells and the river network in the SHUD. (a) water balance in river channels, (b) topologic relationship between river channels and hillslope cells, and (c) water fluxes between river segments and hillslope cells.

volume is formed by the vertical extension of the Delaunay triangles to produce three layers: land surface, unsaturated zone, and groundwater layer. The modeler is responsible for defining the aquifer depth (from the land surface to the impervious bedrock) based on measurements or terrestrial characteristics. The thickness of the unsaturated zone ($D_{us}$) is determined by

the difference between the land surface elevation ($z_{sf}$) and groundwater table ($z_{gw}$) above datum, i.e. $D_{us} = z_{sf} - z_{gw}$. When the groundwater table reaches the land surface ($z_{gw} > z_{sf}$), the unsaturated zone disappears.

Figure 3 depicts the exchange of water between the rivers and hillslope cells. Within each river channel, there are two longitudinal fluxes and two lateral fluxes: upstream ($Q_{up}$), downstream ($Q_{dn}$), overland ($Q_{sf}$) and groundwater ($Q_{sub}$).

The hydrological model solves the Ordinary Differential Equations (ODEs) describing the water state variables using a global implicit numerical solver. The state variables include water height on the land surface ($\boldsymbol{Y_{sf}}$), soil moisture ($\boldsymbol{Y_{us}}$), groundwater gradient ($\boldsymbol{Y_{gw}}$), and river stage ($\boldsymbol{Y_{riv}}$). The initial value problem for these ODEs is formulated as

$$\frac{d\boldsymbol{Y}}{dt} = f(t, \boldsymbol{Y}), \qquad \boldsymbol{Y}(t_0) = \boldsymbol{Y_0},$$

where the discrete state vector is denoted by $\boldsymbol{Y}$,

$$\boldsymbol{Y} = \begin{pmatrix} \boldsymbol{Y_{sf}} \\ \boldsymbol{Y_{us}} \\ \boldsymbol{Y_{gw}} \\ \boldsymbol{Y_{riv}} \end{pmatrix},$$

$\boldsymbol{Y_0}$ are the initial conditions to start the simulation and $f(t, \boldsymbol{Y})$ denotes the equations governing the hydrological flow, which

are described in this section.

The system of ODEs describing the hydrological processes are fully coupled and solved simultaneously at each time step ($\Delta t = t_n - t_{n-1}$) using CVODE, a stiff solver based on Newton-Krylov iteration (Hindmarsh et al., 2019). In brief, the CVODE solver calculates $\boldsymbol{Y}(t_n)$, given $\boldsymbol{Y}(t_{n-1})$ and $\frac{d\boldsymbol{Y}}{dt}|_{t_{n-1}}$. The technical description of the CVODE solver can be found in the literature (Hindmarsh et al., 2019, 2005; Cohen and Hindmarsh, 1996). The kernel governing equations in SHUD are provided

in table 1.

Figure 4 is the workflow within the SHUD model. The explicit model time step (MTS) $\Delta t = t_n - t_{n-1}$ is user-specified, typically varying from one minute to one hour. Within the MTS, both the interception by the vegetation canopy and actual ET are calculated based on prescribed meteorological data, along with calculated soil moisture and groundwater table. The fluxes



**Table 1.** The kernel governing equations in the SHUD model.

| Physical process | Method | Governing equation | Reference equation |
|---|---|---|---|
| Interception | Bucket model | $\frac{dS_{ic}}{dt} = P - E_{ic} - P_{tf}$ | 1 |
| Snow melt | Temperature Index Model | $\frac{dS_{sn}}{dt} = P - E_{sn} - q_{sm}$ | 9 |
| Overland flow | St. Venant Equation (2D) | $\frac{\partial h}{\partial t} + \frac{\partial (uh)}{\partial x} + \frac{\partial (vh)}{\partial y} = q$ | 11 |
| Unsaturated zone | Richards Equation | $C(\psi)\frac{\partial \psi}{\partial t} = \nabla - K(\psi) \cdot \nabla(\psi + Z)$ | 15 |
| Groundwater flow | Richards Equation | $C(\psi)\frac{\partial \psi}{\partial t} = \nabla - K(\psi) \cdot \nabla(\psi + Z)$ | 18 |
| River channel | St. Venant Equation (1D) | $\frac{\partial h}{\partial t} + \frac{\partial (uh)}{\partial x} = q$ | 25 |

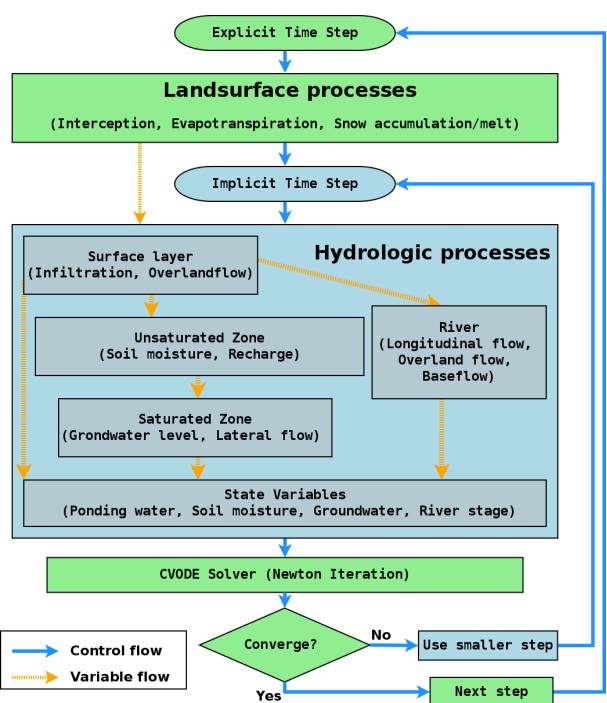

**Figure 4.** The flowchart of calculation of variables in SHUD and time step control.





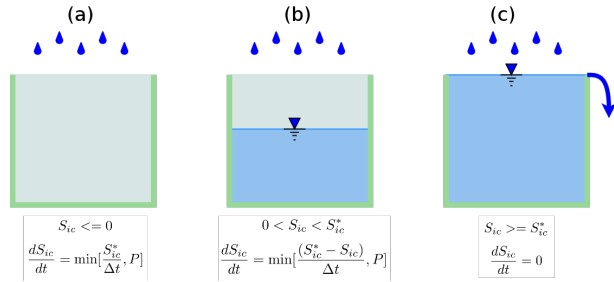

**Figure 5.** The three conditions for the interception calculation within the imaginary canopy *bucket*.

from ET and interception are such slow processes that full coupling of the ET with soil water is not necessary for the model;
instead, within each MTS, the interception, snow accumulation and ET become explicit boundary conditions applied to the
land surface and subsurface. In other words, the interception, ET and snow calculations are synchronized at the MTS, while
the calculation of $Y_{sf}$, $Y_{us}$, $Y_{gw}$ and $Y_{riv}$ use the implicit time step (ITS).

The CVODE solver determines the ITS automatically based on both the specified tolerances and the error function of $Y$ and
$dY$ in CVODE. The initial ITS is set equal to the explicit MTS. Within the ITS, $dY$s is calculated based on $Y$ from the last
MTS. If the CVODE solver converges with the current value of the ITS, it returns the updated $Y$. Otherwise, a convergence
failure occurs that forces an ITS reduction.

The introduction to the mathematical model underlying SHUD is now addressed in five components: vegetation and evapo-
transpiration, land surface, unsaturated layer, saturated layer, and river channel.

### 2.2.1   Vegetation and evapotranspiration

The first calculation performed after receiving atmosphere precipitation is the interception and snow accumulation. The in-
terception is the water loss of precipitation when vegetation cover exists, and is treated as an imaginary *bucket* — namely,
precipitation cannot reach the land surface until the interception bucket is full. The capacity of this bucket is the maximum
interception volume, which is a function of the vegetation Leaf Area Index (LAI) and satisfies equation $S_{ic}^* = C_{ic}LAI$, where
$LAI$ represents the coverage of vegetation canopy over the land area (area of leaves over area of land, $m^2/m^2$), and $C_{ic}$ is
interception coefficient [$m$]. The default $C_{ic}$ is $0.2kg/m^2$ as suggested in Dickinson (1984).

The interception is equal to the deficit of interception – the difference between interception capacity ($S_{ic}^*$) and existing
interception storage ($S_{ic}$). If precipitation is less than the deficit, interception is equal to the precipitation rate (see Fig. 5).

$$\frac{dS_{ic}}{dt} = q_{ic} - E_{ic} \tag{1}$$

$$q_{ic} = \begin{cases} \min[\frac{S_{ic}^*}{\Delta t}, P] & S_{ic} <= 0 \\ \min[\frac{(S_{ic}^* - S_{ic})}{\Delta t}, P] & 0 < S_{ic} < S_{ic}^* \\ 0 & S_{ic} >= S_{ic}^* \end{cases} \tag{2}$$





Potential Evapotranspiration (PET) is the quantity of water that would evaporate and transpire from an ideal surface if extensive free water was available to meet the demand (Maidment, 1993; Kirkham, 2014). As such, PET is a practical and rapid estimation of water flux from land to atmosphere. The PET ($E_0$) is governed by Penman-Monteith equation(Penman, 1948):

$$E_0 = \frac{1}{\lambda} \frac{\Delta(R_n - G) + \rho_a c_p \frac{(e_s - e_a)}{r_a}}{\Delta + \gamma\left(1 + \frac{r_s}{r_a}\right)}.$$ (3)

Here we do not elaborate on this equation, as it is common among different hydrological models (Allen, 1998; Maidment, 1993). At each ET step, the model calculates PET in terms of the prescribed forcing data. PET values are conditioned on the parameters from various land cover types, factored by varying albedo, LAI, and roughness length.

The total Actual Evapotranspiration (AET) consists of three parts: evaporation from interception($E_c$), transpiration from vegetation canopy ($E_t$) and direct evaporation of soil ($E_s$). The calculation of AET for these three components follows from the equations below:

$$
\begin{aligned}
E_c &= \max[S_{ic}/\Delta t, E_0], & (4)\\
E_s &= E_0 \beta_s (1 - \alpha_{imp})(1 - \alpha_{veg}), & (5)\\
E_t &= E_0 \beta_s (1 - \alpha_{imp})\alpha_{veg}, & (6)\\
\beta_s &= \frac{\theta - \theta_r}{\theta_{fc} - \theta_r}. & (7)
\end{aligned}
$$

Here, $E_c$ is subject to PET and the water availability in interception storage. $E_c$ uses water in the interception storage with evaporation rate equal to PET. Both $E_s$ and $E_t$ are affected by soil water stress ($\beta_s$) and impervious area fraction ($\alpha_{imp}$). Impervious area is also considered a barrier of evapotranspiration in the model. $E_s$ is referred to as the demand water evaporation from soil, and emerges from two sources, namely the evaporation from ponding water ($E_{sp}$) and evaporation from soil moisture ($E_{sm}$), i.e. $E_s = E_{sp} + E_{sm}$. The ponding water has higher priority to evaporate — namely, direct evaporation only uses the water in the surface when ponding water is able to meet the $E_s$ demand, i.e. $y_{sf} > E_s$. When ponding water is insufficient to meet $E_s$, soil water balances the difference between demand and available water in the surface; when ponding water does not exist, direct evaporation extracts water from the soil profile ($E_{sm} = E_s, E_{sp} = 0$):

$$
\begin{cases}
E_{sp} = E_s, & E_{sm} = 0, & y_{sf} > E_s \times \Delta t,\\
E_{sp} = y_{sf}/dt, & E_{sm} = E_s - E_{sp}, & y_{sf} < E_s \times \Delta t,\\
E_{sp} = 0, & E_{sm} = E_s, & y_{sf} <= 0.
\end{cases}
$$ (8)

Transpiration also has two potential sources: soil moisture and groundwater from the groundwater table and root depth for the land-use class. Once the groundwater table is higher than the root zone depth, vegetation uses groundwater, and soil moisture stress for transpiration is equal to zero ($\beta_s = 0$).





Water balance associated with snow accumulation is quantified via

$$\frac{dS_{sn}}{dt} = P_{sn} - q_{sn}, \tag{9}$$

$$q_{sn} = (T - T_0) \times m_f, \tag{10}$$

Snow melt rate is determined with snow melt factor ($m_f$), air temperature ($T$) and temperature threshold ($T_0$) at which snow melt occurs. This formulation is often referred to as the degree-day method, in which the values of the snow melt factor and temperature threshold are empirical (Maidment, 1993; Beven, 2012). The water from snow melt is considered as a direct water contribution to the land surface.

### 2.2.2   Water on the land surface

Water balance on the land surface is given by:

$$\frac{dy_{sf}}{dt} = P_n - E_{sp} - q_i - \sum_{j=1}^{3} \frac{Q_s^j}{A_c}, \tag{11}$$

$$P_n = P - S_{ic} + q_{sn}, \tag{12}$$

$$Q_s^j = \frac{L_j}{n} \overline{y_{sf}}^{\frac{5}{3}} s_0^{\frac{1}{2}}, \tag{13}$$

The water balance of net precipitation ($P_n$), infiltration ($q_i$), evaporation from the ponding layer ($E_{sp}$) and horizontal overland

flow ($Q_j$) determine the storage of water on the land surface. Net precipitation ($P_n$) is the total residual water after adjusting for rainfall/snow interception and snowmelt. The overland flow $Q_s^j$ in direction $j$ is calculated with Manning's equation (13). Here $\overline{y_{sf}}$ is effective water height, determined by the gradient between two cells,

$$\overline{y_{sf}} = \begin{cases} y_{sf} & z_{sf} + y_{sf} >= z_{sf}^j + y_{sf}^j \\ y_{sf}^j & z_{sf} + y_{sf} < z_{sf}^j + y_{sf}^j \end{cases} \tag{14}$$

Estimating infiltration utilizes Richards equation,

$$q_i = K_{ei}(\Theta)\left(1 + \frac{y_s}{D_{inf}}\right)$$

$$K_{ei}(\Theta) = \begin{cases} K_r(\Theta)k_x(1-\alpha_h) + \alpha_h k_m \Theta & y_s/\Delta t >= K_{max} \\ K_r(\Theta)k_x(1-\alpha_h) & y_s/\Delta t < K_{max} \end{cases}$$

$$K_r(\Theta) = \Theta^{\frac{1}{2}}\left(-1 + \left(1 - \Theta^{\frac{\beta}{\beta-1}}\right)^{\frac{\beta-1}{\beta}}\right)^2$$

$$K_{max} = k_x(1-\alpha_h) + \alpha_h k_m.$$

The infiltration rate is a function of soil saturation ratio ($\Theta$), soil properties ($k_x$, $k_m$ $\alpha$, $\beta$ and $\alpha_h$) and ponding water height

(existing ponding water plus precipitation/irrigation). Infiltration occurs in the top soil layer ($D_{inf}$), and the infiltration rate is subjected to ponding water height and soil moisture. The default value of $D_{inf}$ is $10cm$, which can be changed in calibration files. The application rate $y_s/\Delta t$ combines ponding water, irrigation and precipitation together, and that determines the





hydraulic gradient applied on the top soil layer. Finally, $K_{max}$ is the infiltration capacity determined by both soil matrix and macropore characteristics. When application rate is less than the maximum infiltration capacity, the infiltration is controlled

by soil matrix flow; when application rate is larger than $K_{max}$, effective conductivity is a function of soil matrix and macropores(Chen and Wagenet, 1992). The infiltration equation takes the macropore effect into account, so the algorithm allows faster infiltration under heavy rainfall events and enables the soil to hold water for vegetation under dry condition.

### 2.2.3 Unsaturated zone

As discussed above, the horizontal flow in the vadose zone is neglected compared to the dominant vertical flow. There are three

processes controlling the water in vadose zone: infiltration ($q_i$), ET in soil moisture ($E_{sm}$) and recharge to groundwater ($q_r$). The calculation of infiltration and ET is explained in the previous subsection. Recharge to groundwater is calculated with the equation 16. The soil moisture content to field capacity controls the recharge rate.

$$s_y \frac{dy_{us}}{dt} = q_i - q_r - E_{sm}, \tag{15}$$

$$q_r = K_{er}\left(\frac{\theta - \theta_r}{\theta_{fc} - \theta_r}\right) \tag{16}$$

$$K_{er} = \frac{D_{us} + y_{gw}}{D_{us}/k_x + y_{gw}/k_v}, \tag{17}$$

Because of the simplification of two-layer description of vertical aquifer profile, we use relationship between soil moisture and field capacity as the gradient to drive the recharge, instead of the hydraulic gradient. $K_{er}$ is the effective conductivity for recharge and is equal to the arithmetic mean of the conductivity of the unsaturated zone and saturated zone.

When the bottom of the vegetation root zone is below the groundwater table, then $E_{tg} > 0$ and vegetation extracts water

from the saturated zone, otherwise $E_{tg} = 0$ meaning that transpiration uses soil moisture. When ponding water exists on the land surface, direct evaporation extracts water from ponding water first; when ponding water is depleted via evaporation, then the remainder of evaporation ($E_{sm}$) uses water from soil moisture based on the water stress.

### 2.2.4 Groundwater

The water balance of groundwater is controlled by the following equations:

$$s_y \frac{dy_{gw}}{dt} = q_r - E_{tg} - \sum_{j=1}^{3} \frac{Q_g^j}{A_c}, \tag{18}$$

$$Q_g^j = \overline{K} \cdot \frac{(y_{gw} + z_b) - (y_{gw}^j + z_b^j)}{d_j} \cdot (L_j \overline{y_{gw}}), \tag{19}$$

$$\overline{K} = (K_{eg} + K_{cg}^j) * 0.5. \tag{20}$$

The calculation of horizontal groundwater flow uses the Richards equation. When the bottom of the root zone is lower than the groundwater table, then $E_t > 0$, otherwise, $E_t = 0$, due to the AET source allocation.

The horizontal groundwater flux $Q_g^j$ is determined by the hydraulic gradient of two adjacent cells, based on the Dupuit-Forchheimer assumption. Above $z_b$ is the elevation of impervious bedrock, $z_b^j$ is the bedrock elevation of its $j$th neighbor cell



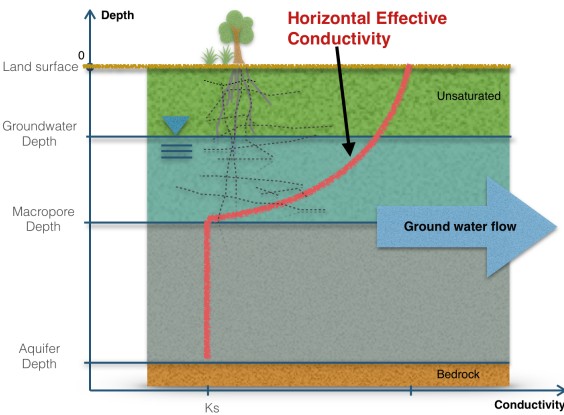

**Figure 6.** Effective conductivity for horizontal groundwater flow changes along the changing groundwater level. When the groundwater level is higher than macropore depth, groundwater flow increases due to the contribution of horizontal macropores.

and $d_j$ is distance between the centroids of two adjcent cells, so the gradient between the two cells is $\left[(y_{gw} + z_b) - (y_{gw}^j + z_b^j)\right] d_j^{-1}$. The effective conductivity for the groundwater flow is the mean value of the effective horizontal conductivity over the two cells. The cross-sectional area along the groundwater flux is equal to $L_j \times \overline{y_{gw}}$.

In equation 20, the effective horizontal conductivity ($K_{eg}$) is a function of the groundwater table and characteristics of the macropores. The calculation of effective horizontal conductivity of each cell is given by

$$K_{eg} = \begin{cases} k_g, & z_m > z_{gw}, \\ \frac{z_{gw} - z_m}{y_{gw}}(k_m \alpha_v + (1 - \alpha_v)k_g) + k_g, & z_m < z_{gw}, \end{cases} \tag{21}$$

$$z_{gw} = y_{gw} + z_{cb}, \tag{22}$$

where $k_g$ and $k_m$ are the saturated hydraulic conductivity of soil matrix and macropores, $z_m$, $z_{gw}$ and $z_{cb}$ are elevations of 380 macropore, groundwater table and bedrock, and $\alpha_v$ is the vertical areal macropore fraction [m$^2$/m$^2$].

     This effective horizontal conductivity can capture increases in saturated flow when the groundwater level rises(Jiang et al., 2009; Bobo et al., 2012; Chen et al., 2018; Cheema, 2015; Taylor, 1960; Lin et al., 2007). Figure 6 reveals the effective horizontal conductivity changes along with different groundwater levels. When the groundwater table is below the level of the macropores, $K_{eg}$ is equal to saturated conductivity. When the groundwater level is above the macropore level, the effective 385 conductivity increases with the groundwater level, taking into consideration the conductivity and area fraction of macropores in the soil profile. The maximum effective conductivity is achieved once the groundwater table level reaches the land surface.





### 2.2.5 Water in streams

The water balance in river channels is described by

$$\frac{dy_{riv}}{dt} = \frac{1}{A_r} \left( \sum_{j=1}^{j=N_c} Q_{sr}^j + \sum_{j=1}^{j=N_c} Q_{gr}^j + \sum_{j=1}^{j=N_u} Q_{up}^j + Q_{dn} \right). \tag{23}$$

The mass balance in each river channel consists of four parts: $Q_s^j$, the overland flow from cells (1 to $N_c$ cells) that intersect with the river channel; $Q_g^j$, the lateral groundwater flux from intersection with the $jth$ cell; $Q_{up}^j$, the longitudinal flow from upstream channels; and $Q_{dn}$, the flux to the downstream channel. $N_u$ is the number of upstream channels; in the model, the number of upstream channels is nonnegative but otherwise unbounded, but only one downstream channel is permitted; namely, we assume river channels can converge into one downstream channel, but cannot bifurcate into multiple downstream channels.

The convergence rule does not affect the topological relationship between river channels and cells.

  The topological relationship between cells and river channels is shown in Fig. 3(b). As depicted, the river consists of a series of river reaches which intersect with the cells. One reach is split as multiple river segments and each segment lies within a hillslope cell. Surface and groundwater exchanges then occur between the segment and the overlay cell. The sum of overland flow from multiple cells contributes to the storage of a river reach.

The downstream channel flux $Q_{dn}$ is based on the one-dimensional diffusive wave equation that is simplified as Manning's equation for open channel:

$$Q_{dn} = \frac{A_{cs}}{\overline{n}} \left( \frac{A_{cs}}{\overline{P}} \right)^{\frac{2}{3}} \overline{s_0}^{\frac{1}{2}}, \tag{24}$$

where $A_{cs}$ is the cross-section area of the river reach, and $\overline{P}$ and $\overline{s_0}$ are average wet perimeter and average slope of a river reach and its downstream reach.

The upstream flux $Q_{up}$ is equal to the sum of $Q_{dn}$ from the multiple upstream reaches. The water balance equation in the river channel neglects evaporation and precipitation because the area of open water in the watershed is relatively small, and the area of open water is already included in pre-computation for the cells. Therefore, the channel routing represents the water exchange between the river and hillslope and takes the overland flow and baseflow into account.

  The overland flow between river segment and associated hillslope cell ($Q_{sr}$) is calculated as follows:

$$
\begin{aligned}
\quad Q_{sr} &= L_s C_w b_s \sqrt{2g|b_s|}, & (25) \\
H_{riv} &= y_{riv} + z_{rb}, & (26) \\
H_{csf} &= y_{sf} + z_{cs}, & (27) \\
b_s &= \begin{cases} H_{riv} - H_{csf}, & H_{riv} > z_{bank} \text{ and } H_{csf} > z_{bank}, \\ H_{riv} - z_{bank}, & H_{riv} > z_{bank} \text{ and } H_{csf} < z_{bank}, \\ H_{csf} - z_{bank}, & H_{riv} < z_{bank} \text{ and } H_{csf} > z_{bank}. \end{cases} & (28)
\end{aligned}
$$

Here $z_{bank}$ is the elevation of riverbank or levee, implying that either the land surface or river stage must be higher than the

levee before water exchange occurs between the land surface and river segment.





The groundwater exchange between river segment and hillslope cell is described by $Q_{gr}$, which is calculated as

$$Q_{gr} = L_s b_g K_{gr} \frac{H_{riv} - H_{cgw}}{d_{rb}}, \tag{29}$$

$$H_{cgw} = y_{gw} + z_{cb}, \tag{30}$$

$$b_g = \begin{cases} y_{riv}, & H_{cgw} < z_{rb}, \\ \frac{1}{2}(y_{riv} + H_{cgw} - z_{rb}), & H_{cgw} > z_{rb}, \end{cases} \tag{31}$$

$$K_{gr} = \frac{1}{2}(k_{rb} + K_{eg}). \tag{32}$$

## 3  Applications

In this section, we present the results of applying SHUD to three hydrological simulations: first, we use the V-catchment experiment to validate the calculation of overland flow and river routing in an idealized catchment; second, we use Vauclin's experiment (Vauclin et al., 1979) to assess the calculation of infiltration, unsaturated flow in the vadose zone and horizontal saturated flow; finally, we apply the model to a hydrological simulation in the Cache Creek Watershed, a headwater catchment in Sacramento Watershed of Northern California.

### 3.1  V-Catchment

The V-Catchment (VC) experiment is a standard test case for numerical hydrological models to validate their performance for overland flow along a hillslope and in the presence of a river channel (Shen and Phanikumar, 2010). The VC domain consists of two inclined planes draining into a sloping channel (Fig. 7). Both hillslopes are $800 \times 1000m$ with Manning's roughness $n = 0.015$. The river channel between the hillslopes is 20 m wide and 1000 m in length with $n = 0.15$. The slope from the ridge to the river channel is 0.05 (in the $x$ direction), and the longitudinal slope (in the $y$ direction) is 0.02.

Rainfall in the VC begins at time zero at a constant rate of $18mm/hr$ and stops after 90 min, producing 27 mm of accumulated precipitation. Since evaporation and infiltration is not involved in this simulation, the total outflow from lateral boundaries and the river outlet must be the same as the total precipitation (following conservation of mass).

Figure 8 illustrates the discharge from the side-plane to the river channel and at the river outlet. The specific discharge (the volume discharge divided by the total area of the catchment) increases with precipitation until it reaches the maximum discharge rate, which is equal to the precipitation rate. Discharges along lateral boundaries and from the river outlet reach the maximum discharge rate, but at different times; namely, the discharge rate from the side-plane reaches the maximum value earlier than in the river outlet. The dots are discharge digitalized from Shen and Phanikumar (2010) with WebPlotDigitizer (https://automeris.io/WebPlotDigitizer/). The results suggest SHUD can correctly capture the processes in overland flow and channel routing, although flow from the river outlet occurs earlier than the prediction in Shen and Phanikumar (2010). Both the fluxes from side-plane and outlet meet the maximum flow rate, that is same magnitude of precipitation after a short period of rainfall. The flux rates start decreasing after precipitation stops. The accumulated volume of flux confirms the correct mass-balance of both fluxes.

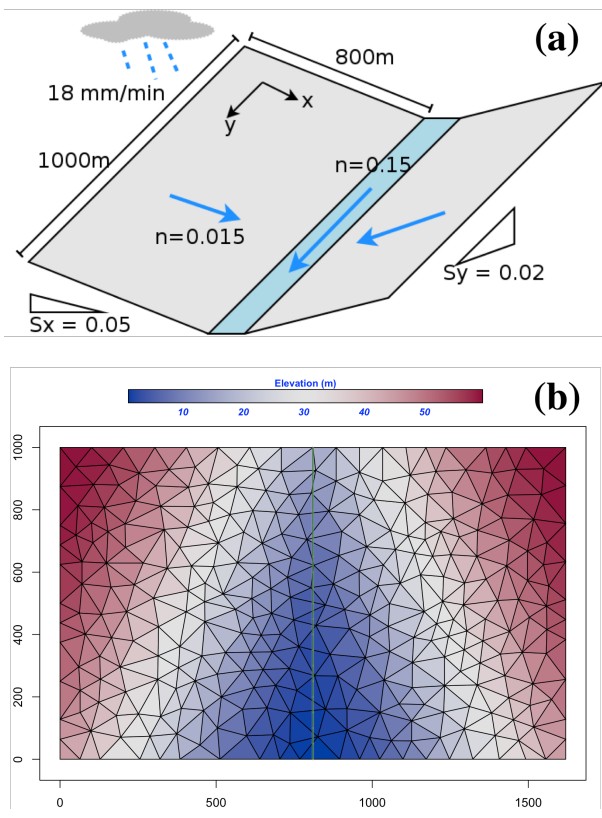

**Figure 7.** The tilted V-Catchment: (a) Basic structure of V-catchment, (b) the SHUD mesh used for the V-Catchment with elevation colored.

In numerical methods, it is necessary to verify the bias of bass-balance on the numerical values within the model — the differences among input, output and storage change in this system (equation 35). The bias in the model result is $\sim 0.2\%$.

$$\Delta S = P - Q - E \qquad (33)$$

$$\hat{\Delta S} = \Delta S_{ic} + \Delta y_{sf} + \Delta y_{us} + \Delta y_{gw} + \Delta y_{riv} \qquad (34)$$

$$Bias = \frac{|\hat{\Delta S} - \Delta S|}{\Delta S} \times 100\% \qquad (35)$$

### 3.2 Vauclin's experiment

Vauclin's experiment (Vauclin et al., 1979) is designed to assess groundwater table change and soil moisture in the unsaturated layer under precipitation or irrigation. The experiment was conducted in a sandbox with dimension 3 m long $\times 2$ m deep $\times 0.05$ m wide (see Fig. 9). The box was filled with uniform sand particles with measured hydraulic parameters: the saturated hydraulic conductivity was 35 cm/hr and porosity was 0.33 m$^3$/m$^3$. The left and bottom of the sandbox were impervious layers, and the



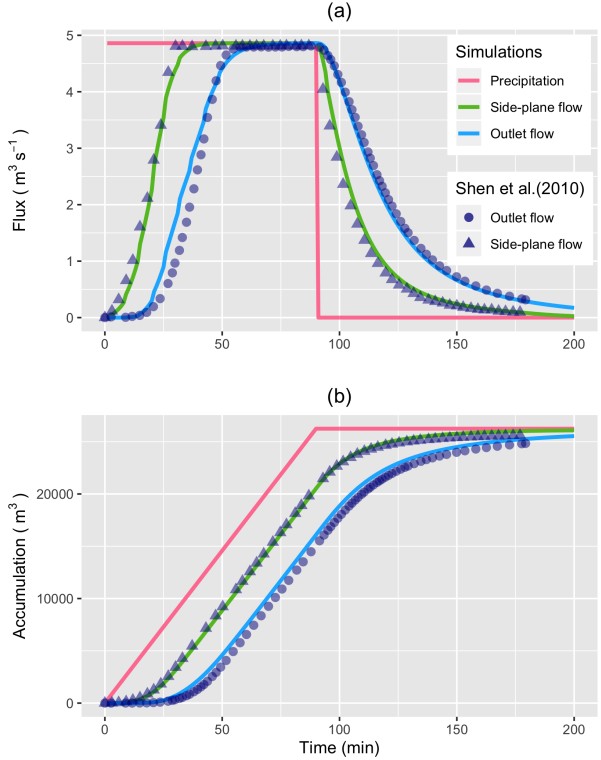

**Figure 8.** Comparison of overland flow and outflow at the outlet of the V-Catchment from the SHUD modeling versus Shen and Phanikumar (2010). (a) is volume fluxes while (b) is accumulated water volume.

top and the right side were open. A hydraulic head was set constant at $0.65m$. Constant irrigation (1.48 cm/hr) was applied over the first 50 cm of the top-left of the sandbox while the rest of the top was covered to avoid water loss via evaporation.

The experiment's initial condition is an equilibrium water table under constant hydraulic head from the right side. That is, the saturated water table across the sandbox was kept stable at $0.65$ m. When the groundwater table reached equilibrium, irrigation was initiated at $t = 0$. The groundwater table was then measured at 2, 4, 6, and 8 hours at several locations along the length of the box. (Vauclin et al., 1979) also use 2-D (vertical and horizontal) numeric model to simulate the soil moisture and groundwater table. The maximum bias between measurement and simulation was $0.52m$, according to the digitalized value of Vauclin et al., 1979, Fig. 10.

Besides the parameters specified in (Vauclin et al., 1979), additional information is needed by the SHUD, including the $\alpha$ and $\beta$ in the van Genutchen equation and residual water content ($\theta_r$). Therefore, we use a calibration tool to estimate the representative values of these parameters. The use of calibration in this simulation is reasonable because the model – inevitably – simplifies the real hydraulic processes. The calibration thus nudges the parameters to *representative* values that approach or fit the *true* natural processes. The calibrated values are $\theta_r = 0.001m^3/m^3$, $\alpha = 0.3$ and $\beta = 5.2$. Like the simulated results in (Vauclin et al., 1979) and (Shen and Phanikumar, 2010), a mismatch exists between the simulations and measurements.



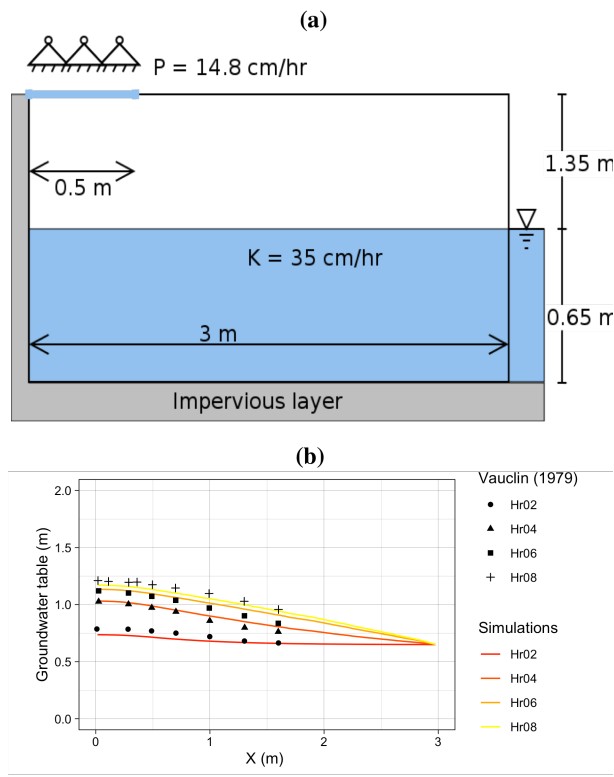

**Figure 9.** A schematic of (a) Vauclin's experiment and (b) a comparison of Vauclin's measurements versus simulated groundwater table change with SHUD.

This mismatch may be due to (1) the aquifer description of unsaturated and saturated layers limiting the capability to simulate infiltration and recharge in the unsaturated zone, or (2) the horizontal unsaturated flow assumptions no longer hold at the relatively microscopic scales of this experiment.

The SHUD simulated the groundwater table at all four measurement points (see Fig. 9(b)). The maximum bias between simulation and Vauclin's observations is $5.5cm$, with $R^2 = 0.99$, that is comparable to the bias $5.2cm$ of numerical simulation in (Vauclin et al., 1979). When the calibration takes more soil parameters into account, the bias in simulation decreases to $3cm$. Certainly, the simplifications employed by SHUD for the unsaturated and saturated zone benefits the computation efficiency while limiting the applicability of the model for micro-scale problems.

The simulations, compared against Vauclin's experiment, validate the algorithm for infiltration, recharge, and lateral groundwater flow. More reliable vertical flow within unsaturated layer requires multiple layers, which is planned in next version of SHUD.



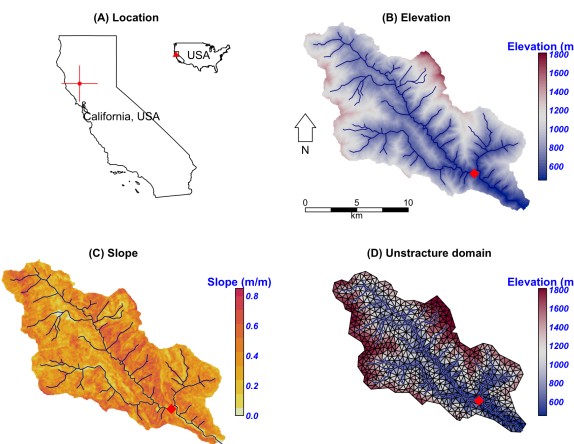

**Figure 10.** The location, terrestrial and hydrological description of the Cache Creek in California. The red diamond in the map is the USGS gage station (11451100) used for calibration and validation.

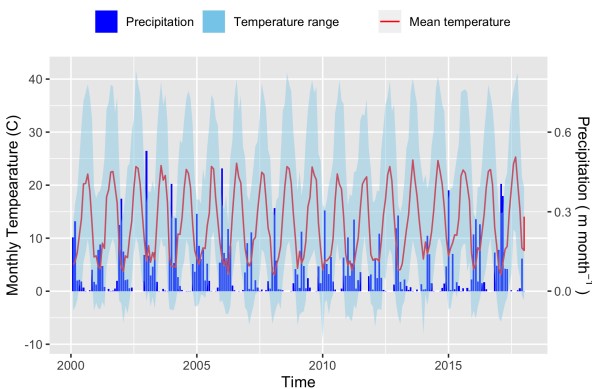

**Figure 11.** The monthly precipitation and temperature in Cache Creek based on NLDAS-2 data from 2000 to 2018. The blue ribbon is monthly precipitation in $m/month$; the red line is monthly mean temperature while blue shadow is the minimum and maximum temperature.

### 3.3 Cache Creek Watershed

The Cache Creek Watershed (CCW) is a headwater catchment with area $196.4km^2$ in the Sacramento Watershed in Northern California (Figures 10 (a), (b) and (c)). The elevation ranges from $450m$ to $1800m$, with a $0.38m/m$ average slope which is very steep, and hence a particularly difficult watershed for hydrologic models to simulate.

According to NLDAS-2, between 2000 and 2017 the mean temperature and precipitation was $12.8°C$ and $\sim 817mm$, respectively, in this catchment. Precipitation is unevenly distributed through the year, with winter and spring precipitation being the vast majority of the contribution to the annual total (Fig. 11).





| Data | Data Source | Type | Resolution |
|---|---|---|---|
| Hydrology | NHD plus(McKay et al., 2012) | Vector | - |
| Elevation | NED(U.S. Geological Survey, 2016) | Raster | 30m |
| Soil | gSSURGO(Soil Survey Staff, 2015) | Vector | - |
| Land-use | NLCD2006(Homer and Fry, 2012) | Raster | 30m |
| Climate | NLDAS-2 FORA(Xia et al., 2012) | Raster | 1/8 deg |

**Table 2.** The basic data sources used to build the model domain of the Cache Creek Watershed.

Table 2 lists the spatial and forcing data supporting the hydrological modeling in CCW. The elevation is 30-meter resolution raster data from National Elevation Dataset(NED)(U.S. Geological Survey, 2016). Forcing data, including precipitation,

temperature, relative humidity, wind speed, and net radiation, is from NLDAS-2 ((Xia et al., 2012) https://ldas.gsfc.nasa.gov/nldas/v2/forcing). Our simulation in CCW covers the period from 2000 to 2007. Because of the Mediterranean climate in this region, the simulation starts in summer to ensure adequate time before the October start to the water year. In our experiment, the first year (2000-06-01 to 2001-06-30) is the spin-up period, the following two years (2001-07-01 to 2003-06-30 ) are the calibration period, and the period from 2003-07-01 to 2007-07-01 is for validation.

The unstructured domain of the CCW (Fig. 10 (d)) is built with SHUDtoolbox, a R package on GitHub ( https://github.com/shud-system/SHUDtoolbox). The number of triangular cells is 1147, with a mean area of $0.17km^2$. The total length of the river network is $126.5km$ and consists of 103 river reaches and in which the highest order of stream is 4. With a calibrated parameter set, the SHUD model tooks 5 hours to simulate 18 years (2000-2017) in the CCW, with a non-parallel configuration (OpenMP is disabled on Mac Pro 2013 Xeon 2.7GHz, 32GB RAM).

Figure 12 reveals the comparison of simulated discharge against the observed discharge at the gage station of USGS 11451100 (https://waterdata.usgs.gov/ca/nwis/uv/?site_no=11451100). The calibration procedure exploits the Covariance Matrix Adaptation – Evolution Strategy (CMA-ES) to calibrate automatically (Hansen, 2016). The calibration program assigns 72 children in each generation and keeps the best child as the seed for next-generation, with limited perturbations. The perturbation for the next generation is generated from the covariance matrix of the previous generation. After 23 generations, the

calibration tool identifies a locally optimal parameter set.

In the calibration period, Nash-Sutcliffe efficiency (NSE Nash and Sutcliffe (1970)), Kling-Gupta Efficiency (KGE, Gupta et al. (2009)) and $R^2$ is 0.72, 0.83 and 0.72 respectively (Fig. 12. The goodness-of-fit in the validation period is less than calibration period (as expected), with NSE = 0.66, KGE = 0.67 and $R^2 = 0.65$. Although the SHUD model captures the flood peaks after rainfall events, the magnitude of high flow in the hydrograph is less than the gage data. There are two potential

causes of this bias: (1) underestimated precipitation intensity from NLDAS-2 data, or (2) over-fitting in the calibration, as the NSE tends to capture the mean value of the observational data rather than the extremes.

Figure 13 represents the monthly water balance in CCW, in which the PET is three times the annual precipitation, but the actual evapotranspiration (AET) is only 27% of the precipitation. This result emerges because the summer is the peak of PET,



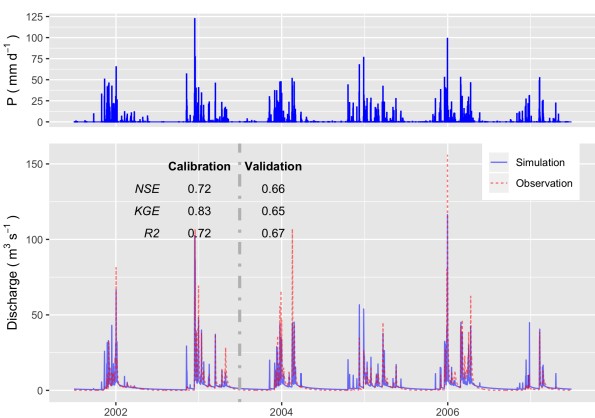

**Figure 12.** The hydrograph in Cache Creek (simulation versus observation) in the calibration (2001-07-01 to 2003-06-30) and validation periods (2003-07-01 to 2007-06-30).

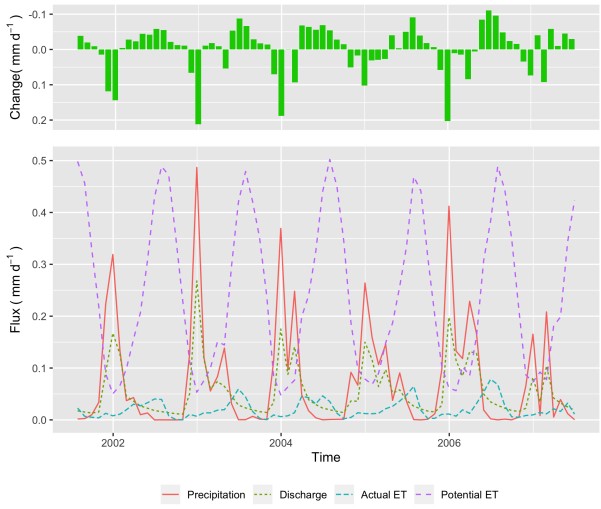

**Figure 13.** The monthly water balance trends in Cache Creek Watershed from 2001-07-01 to 2007-06-30. Top: net change of water storage; Bottom: fluxes of precipitation, actual evapotranspiration, potential evapotranspiration and discharge at the outlet.

while winter is the peak of precipitation and water availability. The AET is subjected to PET and water availability, so the
maximum of AET occurs in early summer. The runoff ratio is about 73%.

We use the groundwater distribution (Fig. 14) to demonstrate the spatial distribution of hydrological metrics calculated from the SHUD model. Figure 14 illustrates the annual mean groundwater table in the validation period. Because the model fixes a $30m$ aquifer, the results represent the groundwater within this 30-meter aquifer only. The groundwater table and elevation along the green line on the upper map are extracted and plotted in the bottom figure. The gray ribbon is the 30-meter aquifer,



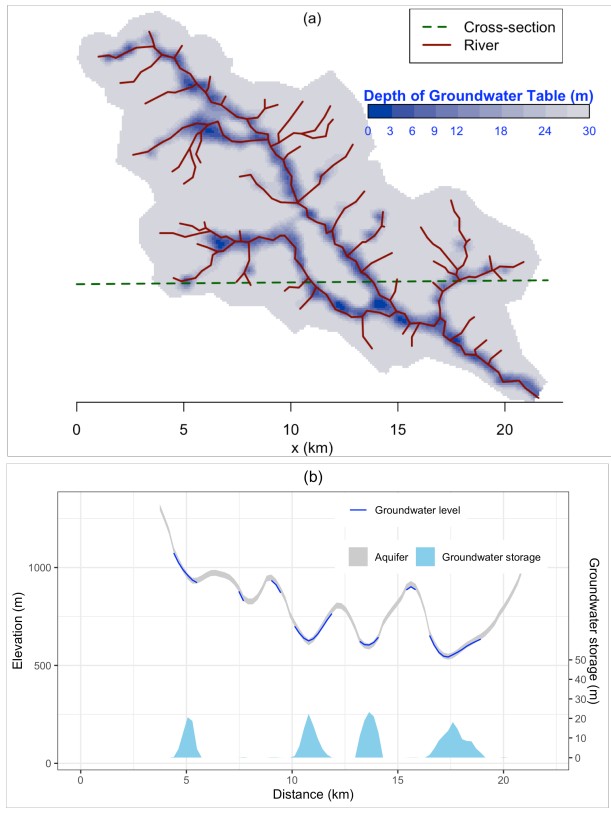

**Figure 14.** groundwater table (top) and the storage of groundwater (bottom) in $30m$ depth aquifer. The groundwater table and elevation along the green line on the top map are extracted and plot in the bottom figure. The gray ribbon is the 30-meter aquifer, and the blue line is the groundwater table, only at the location where groundwater storage is larger than zero. The green polygons with the right axis are the groundwater storage along the cross-section line.

and the blue line is the location where groundwater storage is larger than zero. The green polygons with the right axis are the groundwater storage along the cross-section. The groundwater follows the terrain, with groundwater accumulated in the valley, or along relatively flat plains. In the CCW, the groundwater is very deep or does not stay on the steep slope.

## 4   Summary

We now summarize the formulation and results from SHUD.

- SHUD is a physically-based model, in which all equations used to emerge from the physics behind the hydrological processes within a catchment. The physical model can predict the water in an ungaged water system. SHUD represents the spatial heterogeneity that influences the hydrology of the region. Consequently, it is practical to couple the SHUD model with models from biochemistry, reaction transport, geomorphology, limnology and other related research areas.





- – SHUD is a fully-coupled hydrological model, where the conservative hydrological fluxes are calculated within the same time step. The state variables are the height of ponding water on the land surface, soil moisture, groundwater level, and river stage, while fluxes are infiltration, overland flow, groundwater recharge, lateral groundwater flow, river discharge, and exchange between river and hillslope cells.

- – The global ODE system solved in SHUD integrates all local ODE systems over the domain and solves with a state-of-the-art parallel ODE solver, known as CVODE (Hindmarsh et al., 2005) developed at the Lawrence Livermore National Laboratory.

- – SHUD permits adaptable temporal and spatial resolution. The spatial resolution of the model varies from centimeters to kilometers based on modeling requirements computing resources. The internal time step of the iteration is adjustable and adaptive; it can export the status of a catchment at time-intervals from minutes to days. The flexible spatial and temporal resolution of the model is valuable for community model coupling.

- – SHUD can estimate either a long-term hydrological yield or a single-event flood.

- – SHUD is an open-source model — anyone can access the source code and submit their modifications/improvements.

## 4.1 SHUD and updates from previous versions

As a descendant of PIHM, SHUD inherits the fundamental idea of conceptual structure and solving hydrological variables in CVODE. The code has been completely rewritten in a new programming language, with a new discretization and corresponding improvements to the underlying algorithms, adapting new mathematical schemes and a new user-friendly input/output data format. Although SHUD is forked from PIHM's track, SHUD still inherits the use of CVODE for solving the ODE system but modernizes and extends PIHM's technical and scientific capabilities. The major differences are the following:

1. SHUD is written in C++, an object-oriented programming language with functionality to avoid risky memory leaks from C. Every function in the code has been rewritten, so the functions, algorithm or data structure between SHUD and PIHM are incompatible.

2. SHUD implements a re-design of the calculation of water exchange between hillslope and river. The PIHM defines the river channel as adjacent to bank cells – namely, the river channel shares the edges with bank cells. This design leads to sink problems in cells that share one node with a starting river channel.

3. The mathematical equations used in infiltration, recharge, overland flow and river discharge are different among the two models. This change is so essential that the model results would be different with the same parameter set.

4. SHUD adds mass-balance control within the calculation of each layer of cells and river channels, critical for long-term or micro-scale hydrologic modeling.





5. Either inner data structure or external input/output formats are different. The inner data structure indicates the organization of data, parameters and operations within the program, as well as the strategy to connect the various procedures in the program. The format of input files for SHUD model is upgraded to a series of straightforward and user-friendly formats. The output of SHUD model supports both ASCII and Binary format. Particularly, the binary format is efficient in writing and post-processing.

We now briefly summarize the technical model improvements and technical capabilities of the model, compared to PIHM. This elaboration of the relevant technical features aims to assist future developers and advanced users with model coupling. Compared with PIHM, SHUD ...

– supports the latest implicit Sundial/CVODE solver up to version 5.0.0 (the most recent version at the time of writing),

– supports OpenMP parallel computation,

– redesigns the program with object-oriented programming (C++),

– supports human-readable input/output files and filenames,

– exposes unified functions to handle the time-series data, including forcing, leaf area index, roughness length, boundary conditions and melt factor,

– exports model initial condition at specific intervals that can be used for warm starts of continued simulation,

– automatically checks the range of physical parameters and forcing data,

– adds a debug mode that monitors potential errors in parameters and memory operations.

## 5 Conclusions

The Solver for Hydrologic Unstructured Domain (SHUD) is a multi-process, multi-scale and multi-temporal hydrological model that integrates major hydrological processes and solves the physical hydrological equations with the Finite Volume Method. The governing hydrological equations are solved within an unstructured mesh domain — triangular cells. The variables in the surface, vadose layer, groundwater and river routing are fully coupled together with a very fine time-step. The SHUD uses one-dimensional unsaturated flow and two-dimensional groundwater flow. River channels connect with hillslope via overland flow and baseflow. The model, while using distributed terrestrial characteristics (from climate, land use, soil and geology) and preserving their heterogeneity, supports efficient performance through parallel computation.

SHUD is a robust integrated modeling system that has the potential for providing scientists with new insights into their domains of interest and will benefit the development of coupling approaches and architectures that can incorporate scientific principles. The SHUD modeling system can be used for applications in (1)hydrological studies from hillslope scale to regional scale, (2) water resource and stormwater management, (3) coupling research with related fields, such as limnology, agriculture,



geochemistry, geomorphology, water quality, and ecology, (4) climate change, and (5) land-use change. In summary, SHUD is a valuable scientific tool for any modeling task associating with hydrological responses.

*Code and data availability.* The source code of SHUD model is kept updating at https://github.com/SHUD-System/SHUD. The code and
data used for this page is archived at ZENODO:

SHUD model: 10.5281/zenodo.3561293.

User manual: 10.5281/zenodo.3561293.

V-catchment: 10.5281/zenodo.3566022

Vauclin(1979) experiment: 10.5281/zenodo.3566020

Cache Creek Watershed: 10.5281/zenodo.3566036

*Author contributions.* Lele Shu – Conceptualization, Investigation, Methodology, Software, Validation, Visualization, Writing original draft and editing

Paul Ullrich – Supervision, Investigation, Writing original draft and editing

Christopher Duffy – Supervision, Investigation, Writing original draft and editing

*Competing interests.* Paul Ullrich is a member of the editorial board of the journal

*Acknowledgements.* Authors Shu was supported by California Energy Commission grant "Advanced Statistical-Dynamical Downscaling Methods and Products for California Electrical System" project (award no. EPC-16-063). Co-author Ullrich was supported by the U.S. Department of Energy Regional and Global Climate Modeling Program (RGCM) "An Integrated Evaluation of the Simulated Hydroclimate
System of the Continental US" project (award no. DE-SC0016605)





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
