# Peer review of "Simulator for Hydrologic Unstructured Domain (SHUD v1.0): Numerical modeling of watershed hydrology with the finite volume method"

_Geoscientific Model Development, 2019_

## Short Comment (SC1) · 23 Jan 2020

Dear authors,

in my role as Executive editor of GMD, I would like to bring to your attention our Editorial version 1.2:

https://www.geosci-model-dev.net/12/2215/2019/

This highlights some requirements of papers published in GMD, which is also available on the GMD website in the 'Manuscript Types' section:

http://www.geoscientific-model-development.net/submission/manuscript_types.html

In particular, please note that for your paper, the following requirement has not been met in the Discussions paper:

- "The main paper must give the model name and version number (or other unique identifier) in the title."

Please add a version number for SHUD in the title upon your revised submission to GMD. Yours,

Astrid Kerkweg

---

## Referee Comment (RC1) · Anonymous Referee #1 · 27 Feb 2020

The paper documents the SHUD hydrological model, which is a successor to PIHM (Penn State Integrated Hydrologic Model) first published in 2004. The paper describes the conceptual structure of the model and the mathematical equations behind the different process modules, and it then presents three test cases. The paper is well written, the model components are clearly presented, and the test cases adequately highlight some of the important features of the model.

Some specific comments and suggestions follow.

It is not until well into the paper (section 4.1) that we learn what distinguishes SHUD from PIHM. Barring this information, this paper would be merely a "reference manual"

for a model developed many years ago. It might be a good idea to convey from the outset that SHUD has many improvements and added features relative to PIHM.

Along the same lines, the only reference provided for the PIHM model is a PhD thesis (Qu, 2004). To help the reader appreciate the evolution of the model from PIHM towards SHUD (there was a FIHM model at some point as well, I believe), it would be useful to cite some of the papers that represent key development stages of the modeling framework and significant applications.

The paper is also lacking in citations (and accompanying contextualization with respect to PIHM/SHUD) of physics-based, distributed, integrated, surface-subsurface hydrologic models (ISSHMs) that are perhaps in many ways more similar (relevant) to PIHM than some of the models that are cited in the paper (VIC, HEC, HBV, SWAT, ...).

The model is described as multi-scale but the actual physical scale most suited for application of the model, if there is one, is not really made clear. There are integrated hydrological models for field-scale applications and for continental-scale applications (and everything in between), and each of these models is very different. Where does SHUD fall? (Are the three test cases - a 3 m long sandbox, a v-catchment less than 1 sq.km., and a 200 sq.km. watershed - a reflection of the range of scales most suited to the model?)

Unless it is standard practice for GMD papers, I don't think the (very long) nomenclature is needed for this paper. Describing each variable (and its units) when it first appears should be sufficient.

The paragraph from lines 164 to 169 seems out of place. It can perhaps be merged with the first paragraph of the Intro?

There is a tendency in the paper to justify some of the key assumptions underlying the model as being perfectly reasonable (e.g., lines 211, 214, and 230-231), whereas of course reality is much more complex and some of these assumptions may actually
represent serious limitations of the model. The authors should maybe try to be a bit more nuanced regarding the key assumptions behind the model.

There is missing information for the Bergstrom reference.

The (insanely!) long list of authors for the Bloschl reference is made even longer by repeating the entire list from Duthmann onward.

---

## Referee Comment (RC2) · Anonymous Referee #2 · 27 Feb 2020

The paper introduce the SHUD model, which is a descendant of PIHM (Penn State Integrated Hydrological Model). The paper is very well written and it represents the model structure, mathematical equations employed in the model and examines the model for three different case studies.

Major comments: - The paper specifies changes in the flow simulations and model discretization to be the major improvements to PIHM hydrological model. The authors need to provide information about how they implement the discretization. will the discretization be implemented through the model or using different software such as PIHMgis? This information needs to be provided as a user manual in Github or Zenodo. - The link to the model source code is not really useful for other users since there is not any user manual that explains how to implement the model. Without providing examples and a thorough user manual the model cannot be applied by other users. - One clear disadvantage of the SHUD model relative to Flux-PIHM, which is the PIHM model coupled with Noah-LSM, is using a temperature index approach instead of energy balance for snowmelt estimations. - In SHUD model deep groundwater cannot be considered in subsurface flow simulations while in Flux-PIHM it is, which is another shortcoming to SHUD model that authors tried to justify by assuming most rocks are impermeable, which is not the case in some cases. - Adding irrigation to the simulation is not possible in the PIHM model and based on what the authors mentioned on page 8, line 198, it is possible in SHUD model simulations. If true, authors need to add this to the list of differences between two models and explain in the model user manual how is that possible. - One drawback to the PIHM model was the assumption of homogeneous soil properties within each cell, which is the same in SHUD. - Page 9, Line 230: Authors claim that it is realistic to assume that the water exits the watershed only through stream discharge, considering that the groundwater lateral flow is insignificant and minimal in so many cases, which is not true. -Authors mention that the mathematical equations are different than what used in PIHM such that they produce different results using the same parameters. The difference and how they are "better" than equations that were used in PIHM should be explained. - Flux-PIHM addresses most of the improvements mentioned on page 28 such as checking the range of forcing data, exporting initial condition, supporting human-readable input and output. The authors do not clearly show how the SHUD is better than the current existing versions of PIHM.

Minor comments: - Page 5, Line 95: snowmelt unit could not be m3/s. - Page 5, Line 101 and 102: Two different parameters have the same annotations. - Page 15, Equation 13: Define Lj.

---

## Author Comment (AC1) · 12 Mar 2020

Dear editor, Thank you for this reminder. I will add the model version number into the title in the revision. The version in this paper is v1.0. Thank you.

---

## Author Comment (AC2) · 12 Mar 2020

Thank the reviewer for these helpful comments concerning my manuscript entitled "Solver for Hydrologic Unstructured Domain (SHUD): Numerical modeling of watershed hydrology with the finite volume method". These comments are all valuable and very helpful for revising and improving my paper, as well as the important guiding significance to my research. We have studied comments carefully and have made corrections, which we hope to meet with approval.

Comments by the anonymous reviewer are pasted here in bold font; our answers are given in normal font.

[Figure]

**Comment 1: It is not until well into the paper (section 4.1) that we learn what distinguishes SHUD from PIHM. Barring this information, this paper would be merely a "reference manual" for a model developed many years ago. It might be a good idea to convey from the outset that SHUD has many improvements and added features relative to PIHM.**

I added a brief history of PIHM and relation to the SHUD model, and then explain the differences of the new model with the original PIHM in the last section.

The conceptual structure of the two-state integral-balance model for soil moisture and groundwater dynamics was devised by (Duffy, 1996), in which the partial volumes occupied by unsaturated and saturated moisture storage were integrated directly upon local conservation equation. This two-state integral-balance structure simplified the hydrological dynamics while preserving the natural spatial and temporal scales contributing to runoff response. Brandes et al. (1998) use FEMWATER to realize the numeric experiments of inflow/outflow behavior within a hillslope-stream scheme. In 2004, Qu (2004) embedded the evapotranspiration and river network, and released Penn State Integrated Hydrologic Model (PIHM) v1.0, which was an important milestone integrating the two-state soil moisture-groundwater process with 2-D surface overland and channel flow. Since PIHM v1.0 (Qu, 2004), the PIHM code became a generic, fully-integrated hydrological model applicable to watersheds and river basins. After that, PIHM v2.0 (Kumar et al., 2009; Kumar and Duffy, 2009) enhanced the land surface modeling and adapted the input/output to accept national geospatial soils data. A GIS-tool, PIHMgis(Bhatt et al., 2014) and the Essential Terrestrial Variables Data Server (HydroTerre Leonard and Duffy (2013)) dramatically facilitated rapid the model deployment and applications with PIHM. Because of the sophisticated hydrological modeling and efficient spatial representative of PIHM, various model coupling project initialized. For example, Flux-PIHM coupled the NOAH Land Surface Model into PIHM to calculate more details in energy balance and evapotranspiration (Shi et al., 2015, 2014). Zhang et al. (2016) coupled a landscape evolution model with PIHM

(LE-PIHM). Bao (Bao, 2016; Bao et al., 2017) coupled a reactive transport module with PIHM (RT-PIHM, RT-Flux-PIHM). Flux-PIHM-BGC (Shi et al., 2018) coupled an ecological biogeochemistry code into Flux-PIHM. The Multi-Module PIHM (MM-PIHM) project (https://github.com/PSUmodeling/MM-PIHM) planned to build a uniform repository for all coupled modules. Still, more PIHM coupling projects are ongoing, such as sediments, lakes, crops, etc.. In addition, a finite volume-based integrated hydrologic modeling (FIHM) was developed (Kumar et al., 2009), which used second-order accuracy and solved 2D unsteady overland flow and 3D subsurface flow. Figure 1 shows the family tree of PIHM and SHUD. Every revision/branch received cross-pollination from others. Although PIHM and SHUD share the same fundamental conceptual model for process integration, the input/output and internal algorithms for each process have been completely re-designed to improve the efficiency of the code execution and allowing improved solution speedup and much larger domains at high resolution. Details of differences between them are summarized in the last section of this paper.

*Figure 1 The family tree of PIHM and SHUD. PIHM and SHUD share the same fundamental conceptual model but use different realization. The PIHMgis and SHUTtoolbox are GIS-tools for pre- and post-processing.*

**Comment 2: To help the reader appreciate the evolution of the model from PIHM towards SHUD (there was a FIHM model at some point as well, I believe), it would be useful to cite some of the papers that represent key development stages of the modeling framework and significant applications.**

Thank you for integral-balance the suggestion. I added a paragraph in the first section, briefly describing the history of PIHM and the coupled modules of the PIHM model. That explains the development of PIHM and why I name the new model as SHUD.

**Comment 3: The paper is also lacking in citations (and accompanying contextualization with respect to PIHM/SHUD) of physics-based, distributed, integrated, surface-subsurface hydrologic models (ISSHMs) that are perhaps in many ways**

**more similar (relevant) to PIHM than some of the models that are cited in the paper (VIC, HEC, HBV, SWAT, ...).**

You are right that some of the models (VIC, SWAT, HBV) are different from PIHM-like integrated hydrological models. I cited the inHM, tRIBs, and PAWS are similar integrated hydrological models with coupled numeric methods. PAWS uses the Finite Difference Method. The rRIBS and inHM also use triangular mesh, and they both utilize the Finite Element Method. I plan to make a model comparison of various modeling scenarios to see the differences among them.

**Comment 4: The model is described as multi-scale but the actual physical scale most suited for application of the model, if there is one, is not really made clear.**

The model is applicable from microscale (sandbox) to a regional scale (large basin). An ongoing simulation of SHUD is on Sacramento Watershed with an area of $\sim 700,000km^2$. Namely, the applicable area of the SHUD model ranges from the hillslope scale $\sim 100m^2$ to $10^6km^2$. We are currently advancing the model with HPC applications.

**Comment 5: I don't think the (very long) nomenclature is needed for this paper. Describing each variable (and its units) when it first appears should be sufficient.**

I moved the nomenclature to the appendix. That explains the meaning of symbols and make the paper readable.

**Comment 6: The paragraph from lines 164 to 169 seems out of place. It can perhaps be merged with the first paragraph of the Intro?**

We rephrased this paragraph and merged it into the first paragraph in the revision.

**Comment 7: There is a tendency in the paper to justify some of the key assumptions underlying the model as being perfectly reasonable (e.g., lines 211, 214, and 230-231), whereas of course reality is much more complex and some of**

**these assumptions may actually represent serious limitations of the model. The authors should maybe try to be a bit more nuanced regarding the key assumptions behind the model.**

Thanks for this suggestion. We rephrased the assumptions and gave more practical options upon them. As every model has its own assumptions, we thought it is useful to explicitly explain the assumptions rather than users summarize based on the equations, simulations, or codes.

**Comment 8: There is missing information for the Bergstrom reference.**

This reference is a technical report. I added the publisher.

**Comment 9: The (insanely!) long list of authors for the Bloschl reference is made even longer by repeating the entire list from Duthmann onward.**

That is true. That is a very long name list. I change the name list to "Bloschl, Günter and Bierkens, Marc F.P. and et. al.", which makes a shorter list.
* * *
[Figure]

Conceptual Integrated Model
(Duffy 1996, Brandes 1998, Qu 2004)

PIHM (Qu 2004)

SHUD

Tool

Support

SHUDtoolbox

PIHMgis

Support

PIHM v1.0

PIHM v2.0

FLUX-PIHM

MM-PIHM

RT-PIHM

LE-PIHM

BGC-PIHM

FIHM

**Fig. 1.** The family tree of PIHM and SHUD. PIHM and SHUD share the same fundamental conceptual model but use different realization. The PIHMgis and SHUTtoolbox are GIS-tools for pre- and post-processing.

---

## Author Comment (AC3) · 12 Mar 2020

Thank the reviewer for these valuable comments concerning the manuscript entitled "Solver for Hydrologic Unstructured Domain (SHUD): Numerical modeling of watershed hydrology with the finite volume method". These comments are very helpful for revising and improving my paper, as well as the important guiding significance to my researches. We have studied comments carefully and have made corrections, which we hope to meet with approval.

*Comments by the anonymous reviewer are pasted here in bold font; our answers are given in normal font.

**Comment 1: The authors need to provide information about how they implement the discretization. Will the discretization be implemented through the model or using different software such as PIHMgis?**

SHUDtoolBox is the tool for pre- and post-process input/output data for the SHUD model. Within the SHUDtoolbox, we used the triangulation program written by Shewchuk (Shewchuk 1996) to generate the Delaunay triangulation. Nevertheless, the SHUD model does not limit the triangulation to Delaunay. Besides the SHUDtoolbox, any GIS tool or language (Matlab, R, Python) has various methods to build a triangular mesh that is acceptable in the SHUD modeling system.

PIHMgis (with supports of C++, Qt and Qgis) is GIS-tool supporting PIHM v2.x, that helps users to build the PIHM model. SHUDtoolbox developed in R, realized GIS functions and data processing for SHUD model.

I added the explanations into the revision. The details of SHUDtoolbox would be another paper.

**Comment 2: The link to the model source code is not really useful for other users since there is not any user manual that explains how to implement the model.**

I updated the Readme file of the model source code and three examples that give direct guidelines to run the model.

**Comment 3: One clear disadvantage of the SHUD model relative to Flux-PIHM, which is the PIHM model coupled with Noah-LSM, is using a temperature index approach instead of energy balance for snowmelt estimations.**

Yes, Flux-PIHM couples the NOAH-LSM into the PIHM model that makes the model more capability on computing snow dynamics. Flux-PIHM is one of the fruitful branches in the PIHM family.

Both PIHM and SHUD, however, are aiming to build a community model that encourages experts from other fields to contribute the model coupling based on their require-

ments, instead of making a sophisticated but clumsy model for all users. ET and snow are important for water balance in a hydrological model and the simple methods are useful for many and even most hydrological model applications. When users think the simple process algorithm cannot meet their requirements, the open-source and simplified design of the direct coupling in PIHM and SHUD allow users to modify specific processes and import their own code in a relatively short time. The simplicity of adding processes is the strength of the PIHM-SHUD line of models.

**Comment 4: In SHUD model deep groundwater cannot be considered in subsurface flow simulations while in Flux-PIHM it is, which is another shortcoming to SHUD model that authors tried to justify by assuming most rocks are impermeable, which is not the case in some cases.**

I rephrased the assumption about the impermeable bedrock in the revision. However, the assumption this version of SHUD makes is that there is an effective depth within which flows contribute to local streams. When this assumption is not reasonable the SHUD model is not appropriate. The impermeable bedrock is however a general assumption utilized in many hydrological models, even though they do not explicitly elaborate on the assumption, such as TopModel, VIC, WRF-Hydro, and SWAT when we use the mass balance equation: **Storage change = Rainfall - ET - Discharge**, and in the long-term period, the storage change is considered to be closing to zero, we already assumed a close boundary and impermeable bedrock.

Indeed, the impermeable bedrock does not apply to all regions or modeling purposes. SHUD has one option to solve this issue — define the exchange of shallow and deep groundwater as time-series boundary conditions, then the influence of the porous bottom boundary is considered in the calculation. However, modeler must pay attention that both the permeable bedrock or bottom boundary conditions raise more uncertainties in the model.

**Comment 5: Adding irrigation to the simulation is not possible in the PIHM model**

**and based on what the authors mentioned on page 8, line 198, it is possible in SHUD model simulations. If true, authors need to add this to the list of differences between two models and explain in the model user manual how is that possible.**

Both PIHM and SHUD use the same algorithms to consider the irrigation. There are two options to embed the irrigation.

1. To preprocess the time-series irrigation as precipitation. This is simplest way, but the model would calculate the interception based on the vegetation features.

2. To apply the irrigation as surface boundary conditions.

**Comment 6: One drawback to the PIHM model was the assumption of homogeneous soil properties within each cell, which is the same in SHUD.** Indeed, it is homogeneous within each cell. SHUD and PIHM allow a surface soil layer (user specified) and a deeper hydrologic layer so it is not quite homogeneous in the vertical. It also is heterogenous areally as each prismatic element receives a separate set of soil properties, In the case of SWAT it uses the HRU idea where heterogeneity exists only between HRU's. Namely, only one set of parameters, including the landuse, soil characteristics, slope of the terrain, lag time, and so on, exist within an HRU. The heterogeneity of distributed model is represented within the differences among computing units (HRUs, elements, cells, and volumes ) all over the domain.

The soil properties in SHUD also vary along the vertical direction due to the macropore effect, where the macropore depth is set by the user. This also impacts the effective conductivity as groundwater levels vary in time.

**Comment 7: Page 9, Line 230: Authors claim that it is realistic to assume that the water exits the watershed only through stream discharge, considering that the groundwater lateral flow is insignificant and minimal in so many cases, which is not true.**

I rephrase the words about this assumption in revision. This should not be an assumption, but a default model configuration. Once the default configuration is not acceptable, the user can alter the configuration based on their research areas and requirements. Examples which can be specified in SHUD include applications with internal boundary conditions such as lakes or reservoirs, pumping wells and channel diversions given appropriate geospatial representation in the model set-up.

**Comment 8: Authors mention that the mathematical equations are different than what used in PIHM such that they produce different results using the same parameters. The difference and how they are "better" than equations that were used in PIHM should be explained.**

The reviewer points out an inconsistency in the explanation of the process equations. What we meant to say was that approximations to process equations such as infiltration can produce very different results (e.g 1-D Richard's equation versus Green and Ampt). What we have tried to do in SHUD is follow four rules: simplicity of the conceptual-mathematical process, approximations that have community acceptance, numerical efficiency of the process equation (particularly in the numerical models), and the ability to simply replace process model code. As all models require calibration, the nudges to the parameters are opting to fit the simulation to the observation.

Still, it is valuable to make a comprehensive comparison of the outcomes from different process equations allowed in SHUD in a future paper.

**Comment 9: Flux-PIHM addresses most of the improvements mentioned on page 28 such as checking the range of forcing data, exporting initial condition, supporting human-readable input and output. The authors do not clearly show how the SHUD is better than the current existing versions of PIHM.**

Flux-PIHM does make useful technical extensions from PIHM which relate to coupling with sophisticated eco-hydrologic, and geochemical sub-modules. Bt adding states to be solved these extensions are computationally restricted to coarse grids, smaller

scales and limited time periods even where HPC resources can be used. The goal of SHUD is to improve the core hydrologic modeling of hillslope, catchment and river basin scales and to allow very large and/or high resolution hydrologic processes over these domains. In the future we expect SHUD and Flux-PIHM to converge as efficiency improvements are adopted for adding new process equations as part of PIHM-SHUD ecosystem improvements in the future.

**Comment 10: Page 5, Line 95: snowmelt unit could not be m3/s.**

Fixed this typo. It is m/s.

**Comment 11: Page 5, Line 101 and 102: Two different parameters have the same annotations.**

Deleted the Line 101. Thank you.

**Comment 12: Page 15, Equation 13: Define Lj.**

$L_j$ is defined in Nomenclature. I will move the Nomenclature into the appendix.

---

## Author Comment (AC4) · 12 Mar 2020

The name of the model is changed to **Simulator for Hydrologic Unstructured Domain (SHUD)**, instead of "Solver for Hydrologic Unstructured Domain (SHUD)". Since the CVODE is the generic numeric solver used in the model, so "solver" in the model name may confuse users.

The new model name will be represented in the revision.

---

## Author Response (AR2)

Thank Andrew for these helpful comments concerning my manuscript entitled Simulator for Hydrologic Unstructured Domains (SHUD v1.0): numerical modeling of watershed hydrology with the finite volume method. These comments are all valuable and very helpful for revising and improving my paper, as well as the important guiding significance to my research. We have studied comments carefully and have made corrections, which we hope to meet with approval.

Comments by the Andrew are pasted here in bold font; our answers are given in normal font.

**(1) L60 - "a heritage" is a strange use. Perhaps replace with "an intellectual descendant"?**

I update the sentences with your suggestion. Thank you.

**(2) L447, Sect4.1. It seems that PIHM was rewritten in C++ in late 2019; I would suggest including this as a feature of SHUD while noting that the current developers of PIHM clearly agree! (There is not yet a release in this GitHub repo, but acknowledging it with some kind of web reference would be good.) https://github.com/shulele/PIHM-4.0**

I am the sole owner and developer in PIHM 4.0 (PIHM++) and SHUD. Both the PIHM++ and PIHM 4.0 were tentative names for the model. In case of confusion, I just closed the PIHM 4.0 project on GitHub.

**(3) Sect4.1. "user-friendly": Is this just the ability to input ASCII or binary? Or does it include something else about how to pass those files to the model? If the former, I would suggest "flexible" rather than "user-friendly".**

"Flexible" is the right words to describe the model. ASCII format input/output data is self-explained and readable for model users. The data structure of the model code design helps model developers to understand and couple it with other modules.

**(4) L448-449. "Every function in the code has been rewritten, so the functions, algorithm or data structure between SHUD and PIHM are incompatible." (a) I believe that you mean "and" (b) Rewriting every function does not necessarily make code incompatible, if the interfaces to the functions stay the same. Could you add the necessary logic to this?**

(a) It should be "and". (b) Most of the functions in SHUD do not exist in PIHM, which are newly defined because of the brand-new design of the SHUD model. Only a few functions of physical equations, such as Manning's Equation and van Genutchen's Equation, are shared in SHUD and PIHM. So SHUD and PIHM follow similar fundamental perceptual model, but different mathematical and computational strategies.

**(5) L455. Please elaborate "has been greatly enhanced" with specific examples (and perhaps less superlative language unless you find it to be warranted)**

Because of a large number of iterations and data passing in the model, the consistency and efficiency within the model are critical. The common computations in different functions within various processes are extracted and defined as shared inline functions, which maintain the consistency of calculation and facilitate the code updating. The elimination of the redundant variables and functions also advance consistency and efficiency.

**(6) L467. Reword: as a computer model, SHUD is incapable of redesigning anything.**

The line should be: SHUD is a program with object-oriented programming (C++).

**(7) L469. Could you explain what you mean by "unified functions"? Do you mean functions with standardized I/O to make it easier for the user to intuit how to interact with SHUD?**

The spreadsheet is the standard format for any Time-Series Data (TSD). The TSD in PIHM is very complicated and full of unnecessarily redundant information. I used to write a program to help PIHM users to convert the spreadsheet data to PIHM format.

**(8) L474-475. There must be some way of referencing these R scripts. Is it possible to create a doi-enabled release of https://github.com/SHUD-System/SHUDtoolbox to Zenodo so this can be properly referenced with a doi?**

I added the DOI of the SHUDtoolbox (DOI:10.5281/zenodo.3758097) into Zenodo and the paper .

**(9) GMD editorial staff will check this too – but so you know: EGU journals require in-text citations to code. Therefore, you should cite yourselves inline as appropriate and add the appropriate references to the reference list.**

Sure, I will pay attention to the in-text citations.

**(10) I will not request that you do this, but if you have any information on compute time or benchmarks against PIHM, this could add to your agruments for why you found it necessary to write SHUD.**

I do have some experiences about the compute time between them but did not compare them on benchmarks yet. I planed to do that later. In my experience, the performance of SHUD is 5 to 20 times faster than the PIHM on the same-magnitude problem (similar number of cells on a watershed).

Besides the changes suggested by the editor, we also made some updates in the third version:

1. The name of the model is changed to *Simulator of Hydrologic Unstructured Domains* — a plural *s* at the end of the *Domain*.

2. Use *hydrologic* in the paper, instead of mixed *hydrologic* and *hydrological*.

3. Add paragraphs that discuss the future work of the model since some questions are unanswered in this paper.

4. Minor updates of language.

[revised manuscript text omitted]

---

## Author Response (AR3)

Reply to the editor's comments:

Dear editor,

Thank you for your editorial comments. Here is the reply to your comments.

**(1) Please use your Zenodo dois to create items for the reference list that you cite throughout the document in all relevant places in which you mention material in online code repositories. I know it may seem embarrassing to cite yourself so often, but this is indeed what EGU journals would like. Indeed, if you do not do this now, you will be asked to do so later. As an example of a paper of mine in which I had to do this quite a lot: https://www.hydrol-earth-syst-sci.net/23/2065/2019/.**

I misunderstood the requirement before. I updated the references as per your requirements in the latest revision.

**(2) Please consider moving your "future work" to its own section, or perhaps not including it. I can see its use, but I am afraid that it weakens and dilutes the conclusions. Your co-authors may have insights about this.**

I remove the "future work" paragraphs.

**(3) Please have one of your native-English-speaking co-authors review the text for English-language usage. I could fully understand your meaning – despite not being a trained hydrologist myself – and that's very encouraging. However, some of your longer additions might be difficult for our non-subject-matter-expert copyediting staff to decipher.**

We read through the paper again. I hope this version makes everything clear. Thank you.